# Development of the Spanish Version of Sniffin’s Sticks Olfactory Identification Test: Normative Data and Validity of Parallel Measures

**DOI:** 10.3390/brainsci11020216

**Published:** 2021-02-10

**Authors:** María Luisa Delgado-Losada, Jaime Bouhaben, Alice Helena Delgado-Lima

**Affiliations:** Experimental Psychology, Cognitive Processes and Speech Therapy Department, Faculty of Psychology, Complutense University of Madrid, Campus de Somosaguas, 28223 Pozuelo de Alarcón, Spain; jaimebou96@gmail.com (J.B.); alicedel@ucm.es (A.H.D.-L.)

**Keywords:** Sniffin´ Sticks, olfaction identification, Spanish population, normative data, cultural adaptation, parallel measures

## Abstract

The Sniffin’ Sticks Olfactory Identification Test is a tool for measurement of olfactory performance developed in Germany and validated in several countries. This research aims to develop the Spanish version of the Sniffin’ Sticks Olfactory Identification Test and obtain normative values for the Spanish population. The parameters are free recall and subjective intensity of odorants are included. The influence of possible demographic covariates such as sex, age, smoking, or educational level are analyzed, and the items that best discriminate are studied. In addition, the internal structure validity of the blue and purple versions is studied as a parallel measure, and a cultural adaptation of the purple version is carried out. For this, three independent samples of normosmic healthy volunteers were studied. To obtain normative values, the sample was of 417 participants (18–89 years). For the internal structure validity study of both versions, the sample was 226 (18–70 years), and for familiarity of the purple version, the sample was 75 participants (21–79 years). Results indicated that men and women and smokers and non-smokers perform equally. However, differences were found as age progresses, being more pronounced after 60 years old in all three measurements of the identification test. This research also provides the items that best discriminate in the blue version and a cultural adaptation for the purple version. In conclusion, the Sniffin’ Sticks Odor Identification Test is a suitable tool for olfactory assessment in the Spanish population. The instrument has been expanded with two new scores, and normative data as a function of age are provided. Its parallel version also seems appropriate for testing, as items have been culturally adapted and evidence of internal structure validity for both versions is reported.

## 1. Introduction

Olfactory tests are necessary tools for adequate assessment of olfactory function [1]. The use of olfactory assessment tests has become more relevant in recent years, due to clinical and research findings that indicate the existence of olfactory alterations derived from traumatic injuries as brain injury [2,3], in surgical or medical procedures for the treatment of some diseases such as larynx tumors requiring partial or total laryngectomy [4], treatment with radiotherapy [5], or pathologies with alterations of the sense of smell such as arterial hypertension [6], liver disease [7,8], diabetes mellitus [9,10], rhinitis, sinusitis [11,12], autoimmune diseases [13,14], inflammatory diseases [15], anxiety [16,17], major depression [18,19,20,21], schizophrenia [22], autism [23], and neurodegenerative diseases [24,25], such as frontotemporal dementia [26,27,28], amyotrophic lateral sclerosis [29], Parkinson’s disease [30,31,32,33], or Alzheimer’s disease [33,34,35,36].

There is full consensus that the sense of smell gradually decreases with age, especially after the age of 60 [37,38,39,40,41,42,43], not so when talking about sex, which effects are controversial [44,45,46,47,48,49,50]. There is also no consensus on whether education level has an influence on olfactory assessment tests [51,52,53,54] nor regarding smoking habits [55,56,57].

Olfactory capacity is evaluated through tests that measure threshold and discrimination and identification of odors. The olfactory threshold represents the level of odor detection at low concentration, meaning the least detectable concentrations of odorant that can be perceived, whereas discrimination is the non-verbal distinction of different odors, while identification refers to the ability to name or associate an odor [20,39,42,44].

The best and most widely validated psychophysical test is the Sniffin’ Sticks Olfactory Test (Burghart GmbH, Wedel, Germany), which evaluates the three dimensions or components of olfaction and is used as a daily routine in the clinical practice of otorhinolaryngology assessment in many European countries [58,59,60,61]. 

The Sniffin’ Sticks Olfactory Test was designed and validated over 20 years ago [61]. Since its first publication, versions of the test have been made with modifications of odors [62,63,64], as well as extended [65,66] and abbreviated versions [67,68,69,70]. Furthermore, it has been adapted and validated in Asia [71,72,73], Australia [74], and in various European countries, such as Romania [75], Italy [76], Greece [77], Portugal [78], Holland [79], the United Kingdom [80], Turkey [81], and Denmark [82,83], and recently in Spain by our group [84]. All published studies have demonstrated the usefulness of the Sniffin’ Sticks Olfactory Test to assess olfaction in different cultures and populations around the world.

Several studies indicate that the Sniffin’ Sticks Olfactory Identification Test alone may function as a screening test for olfactory dysfunction or follow-up of olfactory function [85,86], and they are more feasible to apply in clinical practice [68,69]. A lot of evidence of test validity has been obtained in other cultures and languages [58,61,72,73,77,78]. There are studies where the Olfactory Identification Test (just the blue version) has been used for validation of the olfactory performance such as in Arabia [73], Romania [75], or Greece [77]. Adaptation of odorants and distractors in the cultural setting where the test is to be applied is necessary. Odor identification is affected by cultural differences because it is based on the individual’s familiarity with the test odorants and descriptors [77,78]. 

In addition to the various cultural adaptations of the original identification test (blue version) [65,66,68,69,70,73,77,79,87,88], many authors have developed different versions or modifications in various aspects of the test, for example, on the number of odorants used, mostly concluding the usefulness of the reduced versions in clinical practice [70,86,87,88]. Other authors have worked on modifications regarding the number of verbal descriptors of the recognition task, finding that increasing the number of descriptors makes it more difficult to identify odors [89]; in the development of self-administrated versions, the results suggest that odor identification with the Sniffin’ Sticks can also be administered by the subjects themselves [90], or in systems that provide a more accurate interpretation of the results in the subject´s random responses in the odor recognition task [91]. Other studies present modifications of the odorants presentation order and the type of labels (verbal descriptors with or without pictures) finding that scores were not significantly different when the subjects were presented either with verbal descriptors only or with verbal descriptors and pictures [92], but differences were found in performance when including background noise or positive concurrent feedback [93].

An aspect of great interest, considered by a few studies in which the Sniffin´ Stick Olfactory Identification Test has been used, is the perceived subjective intensity with which each odorant is perceived [94,95,96]. Odor intensity assessments performed with the Sniffin’ Sticks are regarded as reliable and valid objective methods [97,98,99]. 

Despite the age-related loss of ability to identify odors being well documented in neurodegenerative diseases and other pathologies, less is known about where the interruption of the process of correct identification occurs. Compared with other measures of olfactory performance, odor identification is a high-level cognitive operation, with greater cognitive load and requires more than the simple ability to perceive odors [100,101]. The process requires semantic knowledge and the ability to retrieve it and to associate the smell retrieved with a linguistic tag. Difficulties at any level of semantic processing can disrupt task performance. Semantic processing is believed to influence odor identification performance, although the degree of influence depends on the format of the odor identification task. In the Sniffin’ Stick Olfactory Identification Test a forced-choice task is performed between a set of verbal labels for odorants. These options are removed if the test is administered in a free response format. Studies have shown that healthy individuals have greater difficulties in labeling odorants in the free response format, exhibiting greater precision in forced-choice formats [102].

As we previously pointed out, evaluation of the capacity for olfactory identification has an increasingly important relevance in both clinical and research contexts [58,59,60]. In both cases, it is often necessary to carry out successive evaluations to assess possible changes: patient follow-up, evolution of deficits, pre- and post-surgical evaluations, evaluation of the efficacy of rehabilitation or the effects of a pharmacological treatment, etc. 

The performance of successive evaluations entails interpretation problems due to the effect of practice, that is, the observation of improvements in performance that are due to previous experience with the test and not to a real change in the patient’s ability. 

As far as we have information, there are no studies using the adaptation of the purple version of the Sniffin´ Sticks Olfactory Identification Test, developed about six years ago by Burghart as an equivalent for the blue version. Having an adaptation in the Spanish population for the purple version is a great advantage, because it will help to avoid the effect of verbal learning of odorants. 

The goals of our study are (i) the development of normative data for the Spanish version of the Sniffin’ Sticks Olfactory Identification Test, including adding two new scores (free recall and subjective intensity) to the original test score (recognition score); (ii) the analysis of covariables for each measurement and description of the normative values; (iii) the study of the items that best discriminate across olfactory identification performance, so a potential shortened screening version of the Sniffin’ Sticks Olfactory Identification Test could be developed; (iv) the obtention of evidence of the internal structure validity of the blue and purple versions of the test as parallel measures; (v) the cultural adaptation of the purple version, by measuring the percentage of familiarity for each odor descriptor.

## 2. Materials and Methods

### 2.1. Participants

Three independent samples were acquired for the present study. First, the Odor Identification Test (blue version) was applied to an initial pool of 547 participants (final sample of *n* = 417), aged 18–89. Both versions of the Odor Identification Test (blue and purple versions) were administered to a second sample of 235 participants (final sample of *n* = 226), aged 18–79 (mean age = 49.66, SD = 15.03). Participants of these two samples were enrolled from social media, public advertisements, senior study participants from our research group, and from Hospital Central de la Cruz Roja (Madrid, Spain). Demographic data (sex, age, educational background) and clinical variables (olfactory alterations, presence of COVID-19 diagnosis, along with its symptomatology, allergies, smoking) were also collected in both samples, so eligibility criteria (inclusion and exclusion) might be checked. 

Inclusion criteria were (i) to be 18 years or older, (ii) absence of current otorhinolaryngology alterations, and (iii) compliance with testing procedure. Exclusion criteria included (i) a medical history of olfactory alterations, including nasal polyposis, sinusitis, or prior nasal surgery, (ii) having reported COVID-19 compatible smell symptomatology, (iii) presence of nasal congestion at the moment of test administration or recent upper respiratory tract infection within two weeks, (iv) medication intake with repercussion in olfactory performance (such as some antibiotics, antiepileptics, antithyroids, benzodiazepines, or antiarrhythmics), (v) presence or suspicion of cognitive impairment and/or neurologic or psychiatric dysfunctions, and (vi) pregnancy.

A third independent sample of 83 participants (final sample of *n* = 75), aged 21–79 (mean age = 51.83, SD = 21.23), was asked to complete a questionnaire about familiarity with the odor descriptors from the Odor Identification Test (purple version). These participants were contacted and enrolled via e-mail. The questionnaire also comprised questions about their current olfactory function, including a self-rating olfactory scale from 0 to 10.

### 2.2. Measures and Testing Procedure

The Odor Identification Test is a part of the Sniffin’ Sticks Olfactory Test (Burghart GmbH, Wedel, Germany) [58,61]. This test aims to objectively measure odor identification performance. In the original version, this test gives a unique score, ranging from 0 to 16, which is obtained by the sum of all the correct answers.

The complete set of the Sniffin’ Sticks Olfactory Test only includes the blue version for identification assessment, a parallel version has been developed: the purple version. The chance of measuring the same construct using parallel versions of the same measurement instrument has a clear advantage: retesting allows control for the practice effect. However, evidence of validity is needed in order to consider the purple version as a parallel measure of the blue one. The present study pretends to support evidence of internal structure validity from both versions of the Odor Identification Test. 

The recent or current Spanish adaptation of the olfactory identification test adds two new scores to the original version, following this order of presentation:**Free recall score:** As in a memory task, free recall implies the odor pen is presented and the participant has to guess the odor descriptor, without alternatives, doing their best to identify the odor descriptor. In a free recall test, information is obtained from what a person is able to remember spontaneously, without the help of clues. This method also requires a major memory component, combined with smell identification. This score is obtained as the total of correct answers from the 16 items, when presented under free recall modality. **Recognition score:** This is the score proposed by the original version [58]. The Odor Identification Test was adapted to the Spanish population by measuring the grade of familiarity with the odor descriptors [84]. The odor pen is presented to the participant and he or she has to recognize the target odor between four odor descriptors. Therefore, this score is obtained by a four-alternative forced-choice method. Correct answers from the 16 items are added in order to calculate this score.**Subjective intensity score:** This score intends to give a subjective measure of odor identification regarding intensity for each pen. This score gives additional value to the test, as it is combined with the other measures of identification performance (free recall and recognition). The subjective intensity score is computed as the arithmetic mean of the intensity given to each item.

These three scores combined allow a profile of odor identification performance to be generated, which also aims to cover memory aspects of odor identification.

Testing procedure is based on a memory task: first, each odor pen is shown to the participant for 3 s, approximately. Then, free recall is required: the participant is asked to recall the odor descriptor. Whether the participant´s answer is the correct, erroneous, or he or she is unable to give an answer, the process moves onto the recognition task. The corresponding card is shown with four alternatives, where only one is correct, and the participant is asked to identify/select the descriptor that corresponds to the odor presented. In both tasks the answer is recorded. This procedure is similar to other memory tests, such as the Word List subtest from Wechsler Memory Scale-IV [103], where free recall is questioned in the first place, followed by recognition. After free recall and recognition tasks, the participant is asked to rate the odor intensity on a scale from 0 (no intensity) to 10 (maximum intensity). This measurement scale for subjective odor intensity was chosen because it is easier to understand for any kind of participant, regardless of age or educational background. For an in-depth description of odor presentation, see Delgado-Losada et al. [84].

Olfactory function was assessed for both nostrils together. For odor presentation, pens with a length of 14 cm and a diameter of 1.3 cm were used. Each pen was filled with 4 mL of the corresponding liquid odorant. The evaluator took the pen’s cap off and put the tip of the pen in front of the participant’s nostrils, with an approximate distance of 2 cm. In any case, the tip of the pen never physically touched the participant’s nose. The overall time of administration ranged from 15 min.

Testing of participants was performed in a quiet, well-ventilated room to avoid any background smell interfering with the test odors and with the use of odorless gloves. All participants were told not to eat, drink, smoke, chew gum, put on cologne, or brush their teeth up to 1 h before participating in the test (they could drink water).

Both the blue and purple versions of the Odor Identification Test were administered to the same participants in a short and similar time interval, using the same evaluator under the same environmental situation. The two versions were presented in two different sessions, with an interval between them of 7 to 10 days. Both versions are similar in content, format and instructions: the same number, type, difficulty, and time of application of the odorant pens. A counterbalance was made in terms of the order of presentation, that is, while approximately half of participants began with the blue version, the other half began with the purple version. The allocation of participants to both groups was random.

### 2.3. Experiment Design

The study was ruled by the principles of the Declaration of Helsinki (Edinburgh, 2013) and was approved by the Ethics Committee from University Hospital San Carlos (Madrid, Spain) (ref. number: 17/192-E). Every participant was told about the study objectives and signed an informed consent prior to measures’ collection. Participants who were online polled agreed with their participation by answering the online survey.

The following experiments composed the study protocol.

**Study 1. Normative data and item analysis for the Spanish administration of Odor Identification Test (blue version).** The Odor Identification Test (blue version), previously adapted to the Spanish population [84], was administered to 547 participants. Due to eligibility criteria (see Section 2.1 Participants) for this initial sample, the final sample was composed of 417 participants (291 females and 125 males) aged from 20 to 84 years (mean age = 58.94, SD = 13.73). Normative data was obtained for this sample (statistics of average, scatter, and position), and item analysis (difficulty index/mean score per item, biserial correlation, and corrected point-biserial correlation as discrimination index) was performed in order to check the quality of the items.

**Study 2. Internal validity of Odor Identification Tests (blue and purple versions).** The theoretical model which underlies the Odor Identification Test assumes the odor identification score is equal to the sum of the correct answers from the 16 items. This implies that the 16 items which compose the test load into a unique factor which intends to measure odor identification. In Study 2, this assumption was checked and evidence of the internal structure validity of the Odor Identification Test was contributed. A sample of 226 participants received both blue and purple versions of the Odor Identification Test. Confirmatory factor analysis (CFA) was applied for each version (blue and purple, independently), establishing a one-factor structure and using the recognition scores. CFA was chosen as the analytical method instead of the exploratory version (exploratory factor analysis, EFA) due to the existence of a prior theoretical model for the Odor Identification Test. Hence, no alternative factor structures should have been checked. Pearson correlation coefficient was later calculated between each factor (blue identification and purple identification). 

**Study 3. Cultural adaptation of Odor Identification Test (purple version).** Odor familiarity with descriptors from the purple version was measured in a sample of 75 participants. This sample was obtained from an online survey (initial sample of *n* = 83), promoted within Complutense University of Madrid. Following the procedure shown in other studies [78,84], participants were asked to rate odor descriptors according to the familiarity degree they thought each odor had with a Likert scale ranging from 1 (not familiar) to 5 (very familiar). Demographic data (sex, age) and olfactory questions (COVID-19 compatible olfactory symptomatology, history of otorhinolaryngology alterations, self-rating of olfactory function from 0 to 10) were also retrieved. Participants with a self-rating olfactory function under 5 were excluded (*n* = 8). This cutoff point was chosen because it covers the range of 5–10 in the scale representing a positive subjective perception of the olfactory performance. Familiarity data was obtained through an online survey; thus, this criterion was selected due to the unavailability of other clinical data. 

The exact translation of the odorants’ descriptors and distractors was done using the established forward-backward procedure. Two independent bilingual (English and Spanish language) health professionals performed translation from English to Spanish language. Two different bilingual health professionals then translated the provisional Spanish version back into English language. The final version was comparable to the original version. As several Spanish translations were found for various odor descriptors, familiarity with these odor descriptors were measured in a Spanish native sample. This procedure follows a similar methodology established by Ribeiro et al. [78] and Delgado-Losada et al. [84].

### 2.4. Statistical Analyses

All statistical analyses were performed with R software, version 3.5.2 [104]. Regarding significance testing, the alpha level was set to 0.05 (α = 0.05).

For Study 1, in the first place, descriptive analysis was performed. After outlier detection and data cleaning (detection and removal of wrong records and records from participants who do not comply with eligibility criteria), the analyzed sample of Study 1 was composed of 417 participants. Later, multiple linear regression analyses were executed on each score (free recall, recognition and subjective intensity) as the dependent variables, including age, sex, smoking status, and educational background as possible predictors. As educational background is a categorical variable with 5 levels (0 = no reading and writing skills, 1 = minimum reading and writing but non-formal learning, 2 = elementary, 3 = secondary education, and 4 = higher education), dichotomous (“*dummy*”) variables pairing each level were introduced in the linear models. Regression coefficients were estimated under the ordinary least squares method. The stepwise procedure was chosen in order to remove non-significant predictors from the regression model. For the descriptive table of Study 1 (Table 1), data were summarized in count, mean, standard deviation, 95% confidence interval of the mean, minimum and maximum, and 5, 10, 25, 50, 75, 90, and 95 percentiles.

Item analysis was performed over the 16 items of blue Odor Identification Test and the three scores (free recall, recognition and subjective intensity), governed by the classical test theory’s principles. For free recall and recognition, dichotomous, difficulty index (proportion of correct answers of each item), point-biserial, and corrected point-biserial correlations were all calculated. On the other side, for subjective intensity, continuous, item mean, point-biserial, and corrected point-biserial correlations were calculated. Corrected point-biserial correlation is interpreted as the discrimination index, as how much the item discriminates between participants’ odor identification performance (i.e., if good smellers or participants with higher olfactory performance are more likely to score the item than participants with lower olfactory performance or bad smellers, it is said that the item is a good discriminant of olfactory function). The cutoff point in this index is traditionally set at 0.2 [105,106]. Items whose discrimination index is below 0.2 are considered to be checked. Items with a discrimination index equal or greater than 0.2 are acceptable, and those equal or greater than 0.3 (but lower than 0.7) are good discriminant items. Cronbach’s alpha was calculated for each score. 

Regarding Study 2, outlier detection and data cleaning were performed. The analyzed sample for Study 2 was composed of 226 participants. This sample underwent two independent confirmatory factor analyses in order to test the theoretical model of the Odor Identification Test. Robust weighted least squares (WLSMV) was picked as the parameter estimation method, as the traditionally chosen maximum likelihood method supposes continuous empirical variables adjusted to a multivariate normal distribution, which is not the case. This method was chosen because it uses tetrachoric correlation for factor extraction (see Flora and Curran [107] for an in-depth explanation of the WLSMV method and its advantages versus the maximum likelihood method with dichotomous empirical variables). Factorial load of item 1 in each model was constrained to 1 in order to avoid under-identification issues. With the objective to check model performance, the following indexes and statistics were chosen. First, the χ^2^ statistic was chosen with as many degrees of freedom (*df*) as the difference between the number of distinct elements in the empirical correlation matrix and the number of parameters estimated by the model points. A *p*-value greater than α = 0.05 indicates a proper fit of empirical data to the proposed model. From this statistic comes the χ^2^/*df* ratio, whose value should be lower than 2 in order to interpret that the empirical data fits the model. The root mean squared error of approximation (RMSEA) was also considered. An RMSEA value lower than 0.06 indicates a good fit of empirical data to the model, whereas a value between 0.06 and 0.08 points to a proper fit. Residuals of each model were analyzed with the root square of the average squared residuals (SRMR), i.e., the standardized index SRMR. The cutoff point in this index is the same as for the RMSEA. In addition, the TLI (Tucker-Lewis index) and the CFI (comparative fit index) were also considered. Values greater than 0.9 indicate proper fit, and values greater than 0.95, a good fit. All cutoff points for fit indexes are taken from Hu and Bentler [108]. Comparative fit criteria (AIC and BIC) are not reported, as the WLSMV estimation method does not allow them to be calculated. Then, MacDonald’s omega and Cronbach’s alpha statistics are reported for internal consistency interpretation. 

Finally, Study 3 intended to adapt the odor descriptors of the purple Odor Identification Test to Spanish speakers. Cultural adaptation of these descriptors allows the purple version of the test to be administered to the Spanish population. Mean and standard deviation for age and self-rated olfactory function and female proportion for sex were obtained. After, ratings for each odor descriptor were averaged and transformed to a percentage scale (where 5 from the Likert scale equals a 100% familiarity). This transformation was done in order to enhance the interpretation of the results, as was also performed by Ribeiro et al. [78] and Delgado-Losada et al. [84]. The cutoff point of 75% familiarity covers Likert choices 4 (quite familiar) and 5 (very familiar), while scores greater than or equal to 50% familiarity cover choice 3 (familiar). See Delgado-Losada et al. [81] for equal analysis in the blue Odor Identification Test.

## 3. Results

### 3.1. Study 1

Descriptive analysis was performed over the three odor identification scores. Descriptive statistics from this normative sample (Odor Identification Test, blue version) are shown in Table 1. Normative data is provided for the three scores which compose the Spanish adaptation of the test: free recall, recognition (the original score) and subjective intensity. 

Following this, each odor identification score was set as a dependent variable in a multiple linear regression analysis, introducing sex, age, smoking status, and educational background as potential predictors. Regression analyses showed a statistically significant main effect of age in free recall (*r* = −0.202, *b* = −0.034, *p* < 0.0001), recognition (*r* = −0.267, *b* = −0.044, *p* < 0.0001) and subjective intensity (*r* = −0.267, *b* = −0.025, *p* < 0.0001). There was not enough evidence of statistically significant effects of sex, smoking status, or educational background in any pair of comparisons (*p* > 0.05). Hence, these scores were categorized in six age groups: 20 s, 30 s, 40 s, 50 s, 60 s, and plus 70. This categorization agrees with a previous study [84] but expanding the older group. Table 1 also shows descriptive statistics for odor identification performance in the three scores per age group.

Figure 1 shows the graphical representation of the mean scores and confidence intervals per age group.

Results for item analysis are reported in Table 2. For free recall and recognition scores (dichotomous items), the difficulty index (proportion of participants who hit the item) and the point-biserial and corrected point-biserial correlations are shown. For subjective intensity (continuous items), item mean, point-biserial, and corrected point-biserial correlations are shown. Regarding the corrected point-biserial correlation (discrimination index), items 1, 4, 9, 10, 14, 15, and 16 accomplish a discrimination index greater than 0.2 [105,106] in the three scores. In addition, item 3 has a discrimination index greater than 0.2 in the free recall score and items 5, 7, and 11 for the recognition score. All 16 items have a discrimination index greater than 0.4 for the subjective intensity score. Cronbach’s alpha statistics for free recall, recognition, and subjective intensity scores in this sample are 0.62, 0.56, and 0.91, respectively.

### 3.2. Study 2

For Study 2, the original recognition scores were used for independent confirmatory factor analyses per test (blue and purple versions) in the analyzed sample of *n* = 226 (56 males and 170 females, mean age = 49.58, SD = 15.01). A tetrachoric correlation matrix is attached in the Appendix A in order to improve analysis reproducibility.

As the 16 items of both Odor Identification Tests are dichotomous (correct/wrong), robust weighted least squares (WLSMV) was chosen as the extraction method. Absolute and comparative fit indexes for both models may be seen in Table 3. One-factor models show proper goodness of fit to empirical data for both the blue and purple versions, with significance testing (chi-square) without enough evidence to reject the models (*p* > 0.05). RMSEA is below 0.05 in both cases, with the 95% confidence interval touching 0. The SRMR value is slightly greater than 0.05. Although they do not reach the cutoff point of 0.9 [101] in either model, both CFI and TLI indexes show better values in the blue one. Figure 2 shows both models with their respective standardized factorial loads. 

Pearson’s correlation coefficient between blue and purple versions total scores (sum of correct items) is 0.71 (*p* < 0.0001) and between blue and purple versions factor scores (resulted from CFAs) is 0.465 (*p* < 0.0001).

### 3.3. Study 3

Familiarity with the odor descriptors from the purple Odor Identification Test was rated by a pool of 75 participants (15 males and 60 females aged between 21 and 79 years (mean = 51.82, SD = 21.23). A 1–5 Likert-type scale was employed for this. All ratings per item were averaged and transformed to a percentage scale, which aimed to measure the percentage of familiarity. Table 4 shows the percentage of familiarity for each odor descriptor.

Half of the odor descriptors (25/50) show familiarity percentages above 75%, but the familiarity of almost all odor descriptors (48/50) was above 50%. The original odorants contained within the pens were unchanged. However, in light of these familiarity results, some descriptors were replaced by terms more familiar to Spanish speakers: *paprica* (paprika, %familiarity = 46.13) with *pimenton dulce* (%familiarity = 73.87) and *chucrut* (sauerkraut, %familiarity = 50.67) with *coles* (%familiarity = 68). Gooseberry was maintained as *grosella* (%familiarity = 40.8) due to a lack of a more suitable semantic descriptor. None of these odor descriptors were odor targets. 

## 4. Discussion

The Sniffin’ Stick Olfactory Identification Test is a screening test for olfactory dysfunction or follow-up of olfactory function clinically [68,69,85,86]. The normative data for the evaluation of the olfactory identification capacity (blue version) identification subtest of the Sniffin´ Sticks Olfactory Test in the Spanish population are presented.

The normative data presented in the tables are to be used as a guide to estimate the individual olfactory identification capacity in relation to the individual’s age. The normative data of the three scores that make up the validation of the Spanish version of the identification test are free recall, recognition and intensity. The tables allow us to compare the performance of people over 20 years old, assigning a range of deciles compared to their peers of a similar age. The decision about this age categorization by 10 years was made based on the intention to capture olfactory differences across the lifespan, following the same procedures as studies in the area, including our previous work [60,74,84]. The 10th percentile has been used to discriminate between normosmic and hyposmic people [58]. 

Our results showed an increasing ability to identify odors, both in free recall and in recognition, up to the age of 40 years, except for the subjective intensity scale, where the youngest group scored higher than the rest of the groups. This may be related to a cognitive bias in youth, an overestimation of the level of competence above reality (Dunning-Kruger effect) well exposed in various studies [109,110,111,112]. The identification score is inversely correlated with age for all measures. Our results indicate a less efficient performance in all olfactory tests from the age of 40, observing a gradual decrease in all age groups. This decrease in the ability to identify odors related to the aging process has been described in numerous previous studies [37,38,39]. 

One of the objectives of the current study was to develop the Spanish version of the Sniffin’ Sticks Odor Identification Test and to obtain normative data of this population. Within this objective, we give special relevance to the +60 cohort, as, from these results, we could plan future studies which dive deeper into the odor identification performance for these ages. Having reference values of the identification test with the free recall, recognition, and intensity measures will allow an assessment of whether the ability to identify odors in a population is normal or impaired. It might be useful to have normative values for each parameter. Olfactory identification requires semantic knowledge and the ability to retrieve it and to associate the smell retrieved from memory with a linguistic tag. Difficulties at any level of semantic processing can disrupt task performance. Although the deficit in the organization of semantic knowledge in patients with Alzheimer’s disease is known, a hypothesis of a break in the semantic network for odors is suggested [113,114,115]. 

Our group is interested in analyzing deficits in semantic networks in future studies and studying the degree of olfactory identification impairment in each mild cognitive impairment subtype, subjective memory impairment, and early Alzheimer’s dementia and assessing the relationship between olfactory identification and cognitive performance. Olfactory identification ability reflects the functional integrity of the human olfactory system. Its deficit is a potential early clinical marker and predictor for Parkinson’s disease and is also implicated in Alzheimer’s disease [116,117]. 

Regarding sex, no statistically significant differences were found in any of the measures of the Sniffin´ Stick Olfactory Identification Test. The results of this research are consistent with those found in other validation studies in different countries [39,48,58,78,80,81], including the one carried out by our group [84], although others indicate that women perform better in the olfactory test due to hormonal factors, especially from the effect of estrogens in the female olfactory epithelium [60,78,86]. The high proportion of female participants in our sample might also mask potential gender differences. Thus, this result, descriptive statistics per sex and age, is attached in Appendix A. No differences have been found in terms of educational level [33,34,35,36], nor between smokers and non-smokers, in the same way as in other validation studies of the identification test where this condition was also considered among participants [39,60,80]. 

The third objective of this work was to study the items that best discriminate in the Sniffin’ Stick Olfactory Identification Test (blue version) and that could constitute an abbreviated version of the test. The items that best discriminate in the three scores of the Spanish version of the identification test are items 1, 4, 9, 10, 14, 15, and 16, which correspond to the odorants orange, mint, garlic, coffee, rose, anise, and fish, respectively. These seven odorants could constitute an abbreviated version of the blue test of the Spanish version of the Sniffin’ Stick Olfactory Identification Test, in the same line as the abbreviated versions proposed by other authors [68,70,88], that could be useful for identifying patients who should undergo more exhaustive and extensive evaluations of their olfactory capacity. In addition, it avoids saturation of patients’ olfactory system, it decreases their fatigue when performing the test, it improves their general performance, and it reduces the possibility of random responses [91].

The internal validity of the test was studied with confirmatory factor analysis, due to the existence of a previous theoretical model of odor identification. These analyses were performed over the two versions of the test (blue and purple versions), as they are considered parallel measures. Study 2 results sustain the theoretical one-factor structure and validate it in both versions. Results also indicate that the two versions correlate with each other. It is a stable measure and can be used as equivalents [118]. These results are useful for clinical practice and research. Repeated administration of the same assessment instrument can produce a practice effect, obtaining stability or improvement in scores that can be explained only by this effect, but masking a real decrease [119,120]. The equivalence between the blue version and the purple version of the Sniffin´ Stick Olfactory Identification Test allows them to be used as parallel forms in follow-up studies, facilitating the interpretation of the results. 

In order to carry out the previous study, it was necessary to perform the cultural adaptation of the purple version of the Sniffin’ Stick Olfactory Identification Test. The purple version was developed about six years ago, and to our knowledge, there are no cultural adaptations or validation studies in any country in the world, just as there were no studies in which its usefulness as a parallel measure of the blue version was analyzed. 

The tests for evaluating the ability to identify odors have important cultural components [10,39,74,79,80,81,82]. The odorants of a well-validated identification test should be familiar to individuals from each country. Cultural adaptation of smells in terms of linguistic and familiarity aspects is necessary before using the test in a country. 

One of the focuses of this study was trying to solve the difficulties derived from the factors of cultural bias by adapting the descriptors used in the odorants and distractors applied. The original odorants contained in the sticks were not modified but the results obtained in the familiarity survey indicated the need to replace the descriptors paprika and sauerkraut with more common terms in the Spanish language, sweet pepper and cabbage. The gooseberry descriptor was kept, even though it was considered to be of low familiarity, due to the lack of a more adequate semantic descriptor. The modifications made do not imply changes in the form of application of the test.

This investigation aims to contribute towards making normative data from a widely used Odor Identification Test available. Whereas this study has a number of strengths, it also has limitations. Future studies would need to consider replicating this research with a larger number of participants, and it would also be in our interest to balance the proportion between female and male participants in order to have a clearer view of the role that gender plays in olfactory performance. Regarding sociocultural level, it could be the case that some of the subgroups encompass a significant sociocultural heterogeneity, or they may have limited knowledge of or previous exposure to the odors used in the test, and the lack of familiarity may influence the performance obtained for certain odorants and the test among all participants. The application of the test in patients and controls is important to be able to determine the specificity and sensitivity of the test, as well as to evaluate the construct validity using the version culturally adapted to people with a reduced sense of smell. Diagnostic capabilities of potential reduced versions of the test (reduced versions with the most discriminant items, as item analysis in Study 1 has shown) should also be assessed. Finally, the test would benefit from including, in future studies, evidence of external and ecological validity, by correlating these measures with others from similar olfactory instruments.

## 5. Conclusions

The present results obtained in this work constitute an important contribution in the evaluation of olfactory capacity, providing different normative data for each of the age groups. 

The results do not indicate that there is a relationship between smell and sex, between smell and educational level, or between smell and smoking. However, changes in olfactory identification are observed as age progresses, changes are seen after 40 years old and the decrease being more pronounced after the age of 60, in all three measures of identification capacity. 

Other contributions of this research are the extraction of the items which best discriminate in the blue version of the test and that could be considered to be used as a shortened or screening version of the test. In addition, evidence of internal structure validity of both versions of the test (blue and purple) is provided through confirmatory factor analysis. Items from the purple version have also been adapted to the Spanish population, as odor descriptors with lower percentages of familiarity were modified. Having a culturally-adapted, parallel version of the Sniffin’ Sticks Odor Identification Test supposes an important advantage in order to improve the quality of follow-up assessments.

In conclusion, the Sniffin’ Sticks Odor Identification Test is a suitable tool to evaluate olfactory identification ability within the clinical and research environment.

## Figures and Tables

**Figure 1 brainsci-11-00216-f001:**
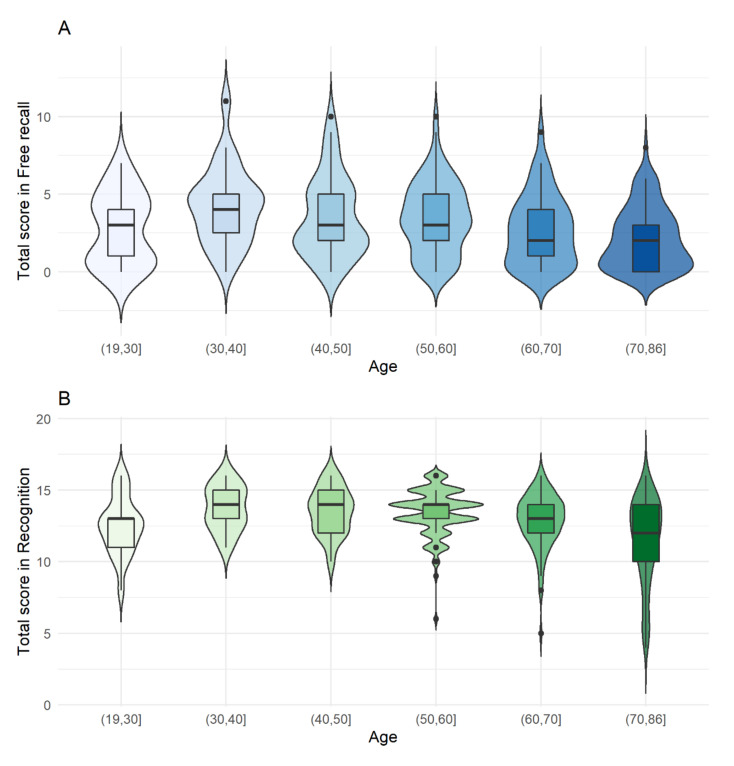
Graphical representation of the blue Odor Identification Test scores per age group: free recall (**A**), recognition (**B**) and subjective intensity (**C**).

**Figure 2 brainsci-11-00216-f002:**
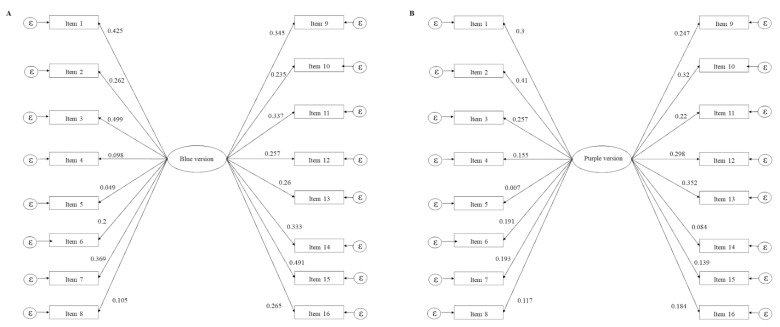
Confirmatory factor analysis models of blue (**A**) and purple (**B**) versions of the Odor Identification Test. For each model, rectangles represent the items (empirical variables) and ovals represent each factor. For each model, all items load into a unique factor, which could be named as the odor identification construct. The standardized factorial load of every item is also reported in each model.

**Table 1 brainsci-11-00216-t001:** Descriptive statistics of free recall, recognition and subjective intensity scores per age group in the Odor Identification Test (blue version).

		Free Recall	Recognition	Intensity
Overall Sample				
*n*		417	417	417
Mean		3	12.86	6.9
SD		2.33	2.18	1.31
Mean CI 95%		[2.78, 3.22]	[12.65, 13.07]	[6.78, 7.03]
Min		0	4	3.06
Max		11	16	10
Percentiles	5	0	9	4.55
	10	0	10	5.01
	25	1	12	6
	50	3	13	7
	75	5	14	7.87
	90	6	15	8.56
	95	7	16	8.87
**Age Group [20–30)**				
*n*		21	21	21
Mean		2.76	12.52	7.83
SD		2.30	2.01	1.23
Mean CI 95%		[1.78, 3.74]	[11.66, 13.38]	[7.3, 8.36]
Min		0	8	4.81
Max		7	16	10
Percentiles	5	0	10	5.87
	10	0	10	6.5
	25	1	11	7.25
	50	3	13	8.31
	75	4	13	8.5
	90	6	15	8.75
	95	6	16	9.75
**Age Group [30–40)**				
*n*		23	23	23
Mean		4.26	13.74	7.21
SD		2.53	1.54	1.09
Mean CI 95%		[3.23, 5.29]	[13.11, 14.37]	[6.77, 7.66]
Min		0	11	4.56
Max		11	16	8.75
Percentiles	5	1	11.1	5.31
	10	1.2	12	5.95
	25	2.5	13	6.5
	50	4	14	7.25
	75	5	15	8.03
	90	7	15.8	8.42
	95	7.9	16	8.55
**Age Group [40–50)**				
*n*		40	40	40
Mean		3.52	13.52	7.33
SD		2.58	1.6	1.10
Mean CI 95%		[2.72, 4.32]	[13.03, 14.02]	[6.99, 7.67]
Min		0	10	4.93
Max		10	16	9.5
Percentiles	5	0	10.95	5.41
	10	1	12	5.74
	25	2	12	6.56
	50	3	14	7.4
	75	5	15	8.25
	90	7.1	16	8.58
	95	8.05	16	8.81
**Age Group [50–60)**				
*n*		139	139	139
Mean		3.47	13.45	6.97
SD		2.30	1.58	1.34
Mean CI 95%		[3.09, 3.85]	[13.19, 13.71]	[6.75, 7.19]
Min		0	6	3.45
Max		10	16	10
Percentiles	5	0	11	4.5
	10	0	11	5.11
	25	2	13	6.22
	50	3	14	7.06
	75	5	14	7.87
	90	6	15	8.62
	95	7	16	9
**Age Group [60, 70)**				
*n*		99	99	99
Mean		2.69	12.87	6.78
SD		2.26	1.88	1.32
Mean CI 95%		[2.24, 3.13]	[12.5, 13.24]	[6.52, 7.04]
Min		0	5	3.94
Max		9	16	9.81
Percentiles	5	0	9.9	4.84
	10	0	11	5.26
	25	1	12	5.78
	50	2	13	6.81
	75	4	14	7.72
	90	6	15	8.59
	95	7	15	8.89
**Age Group [>70)**				
*n*		95	95	95
Mean		2.16	11.58	6.47
SD		1.97	2.94	1.23
Mean CI 95%		[1.76, 2.55]	[11.99, 12.17]	[6.22, 6.71]
Min		0	4	3.06
Max		8	16	9.06
Percentiles	5	0	5	4.05
	10	0	7	4.94
	25	0	10	5.81
	50	2	12	6.62
	75	3	14	7.37
	90	5	15	7.94
	95	6	15	8.19

SD = standard deviation, CI = confidence interval, Min = minimum, Max = Maximum.

**Table 2 brainsci-11-00216-t002:** Item analysis of Odor Identification Test (blue version) with the three scores.

		Free Recall			Recognition			Subjective Intensity		
	Odor Target	Difficulty	Bis	pBis	Difficulty	Bis	pBis	Item Mean	Bis	pBis
Item 1	Orange	0.247	0.281	**0.214**	0.952	0.32	**0.213**	6.863	0.638	0.62
Item 2	Leather	0.05	0.179	0.086	0.815	0.226	0.178	5.89	0.584	0.57
Item 3	Cinnamon	0.271	0.389	**0.305**	0.765	0.2	0.16	6.6	0.583	0.562
Item 4	Mint	0.415	0.365	**0.299**	0.906	0.45	**0.353**	7.758	0.665	0.642
Item 5	Banana	0.264	0.375	0.199	0.923	0.353	**0.267**	7.297	0.664	0.624
Item 6	Lemon	0.072	0.313	0.177	0.575	0.161	0.129	5.642	0.63	0.616
Item 7	Liquorice	0.122	0.295	0.196	0.818	0.354	**0.285**	6.65	0.65	0.335
Item 8	Solvent	0.014	0.37	0.123	0.592	0.029	0.023	6.68	0.648	0.626
Item 9	Garlic	0.35	0.448	**0.367**	0.858	0.352	**0.286**	8.024	0.702	0.684
Item 10	Coffee	0.367	0.429	**0.348**	0.868	0.328	**0.256**	7.218	0.685	0.673
Item 11	Apple	0.029	0.299	0.127	0.52	0.33	**0.249**	6.369	0.625	0.612
Item 12	Clove	0.053	0.354	0.194	0.729	0.176	0.141	6.774	0.641	0.626
Item 13	Pineapple	0.019	0.356	0.138	0.791	0.215	0.169	6.376	0.68	0.663
Item 14	Rose	0.129	0.402	**0.271**	0.892	0.376	**0.29**	7.141	0.695	0.677
Item 15	Anise	0.216	0.373	**0.282**	0.9	0.41	**0.323**	6.914	0.642	0.624
Item 16	Fish	0.381	0.41	**0.335**	0.966	0.34	**0.222**	8.247	0.639	0.62

Discrimination indexes over 0.2 are highlighted in bold.

**Table 3 brainsci-11-00216-t003:** Goodness of fit for confirmatory factor analysis (CFA) in blue and purple versions of the Odor Identification Test.

Absolute Fit Indexes	Blue	Purple
$\chi^{2}$, *df* (*p* value)	116.277, 104 (0.193)	114.114, 104 (0.234)
$\chi^{2}$/*df*	1.12	1.10
RMSEA [CI 95%]	0.023 [0, 0.042]	0.021 [0, 0.041]
SRMR	0.062	0.061
CFI	0.831	0.756
TLI	0.805	0.719
**Internal Consistency Statistics**		
MacDonald’s omega	0.57	0.43
Cronbach’s alpha	0.6	0.45

**Table 4 brainsci-11-00216-t004:** Familiarity percentage of purple Odor Identification Test’s descriptors.

Original Odor Descriptor	Proposed Spanish Translation	% Familiarity	Original Odor Descriptor	Proposed Spanish Translation	% Familiarity
Coffee	Cafe	96.80	Caramel	Caramelo	74.40
Orange	Naranja	95.73	Parsley	Perejil	74.13
Garlic	Ajo	93.33	Paprika	Pimenton dulce	73.87
Chocolate	Chocolate	92.27	Salami	Salami	73.07
Lemon	Limon	91.47	Carrot	Zanahoria	71.20
Ham	Jamon	90.67	Peanut	Cacahuete	70.93
Onion	Cebolla	90.67	Mustard	Mostaza	70.13
Cinnamon	Canela	89.07	Gummy	Gominola	69.87
Grass	Cesped	88.80	Coke	CocaCola	69.60
Rose	Rosa	87.73	Smoked meat	Carne ahumada	68.27
Eucalyptus	Eucalipto	86.93	Sauerkraut	Coles	68.00
Strawberry	Fresa	85.60	Mushroom	Champiñon	68.00
Cigarette	Tabaco	85.60	Grape	Uva	67.47
Melon	Melon	84.80	Lilac	Lila	67.20
Apple	Manzana	84.00	Nutmeg	Nuez moscada	65.33
Lavender	Lavanda	83.73	Raspberry	Frambuesa	65.07
Peach	Melocoton	81.60	Cherry	Cereza	63.20
Wood	Madera	81.33	Ginger	Jengibre	62.40
Coconut	Coco	80.53	Fir	Abeto	61.87
Liquorice	Regaliz	78.67	Plum	Ciruela	61.07
Vanilla	Vainilla	78.13	Chive	Cebollino	58.67
Leather	Cuero	77.07	Grapefruit	Pomelo	55.20
Cookies	Galletas	76.27	Sauerkraut	Chucrut	50.67
Pepper	Pimienta	75.73	Paprika	Paprica	46.13
Pear	Pera	74.4	Gooseberry	Grosella	40.80

## Data Availability

Data at individual level is available upon request to first author.

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
