# Peer review of "Development of the Spanish Version of Sniffin’s Sticks Olfactory Identification Test: Normative Data and Validity of Parallel Measures"

_brainsci, 2021, doi:10.3390/brainsci11020216_

Round 1

Reviewer 1 Report

Overall, a well-researched article, however several changes must be made. 

Language: Please re-check, there are still some errors

I have several concerns:

  • the aims discussed are not all mentioned in the abstract or the introduction, e.g. "the aim of this study was to obtain normative data for the group of people over 60" --> not mentioned in the abstract, this also applies to the "third objective"
  • please adapt the abstract to clearly state the aims so that that is is well rounded (conclusion also doesn't fit the abstract)
  • In materials and methods you write that in any  case, the tip of the pen physically touched the participants nose. Could this be a miscommunication? Did you meant to write that in no case did it touch the nose?
  • please include more information on the free recall score in the methods (what exactly the possible scores meant)
  • How was the Odour identification test adapted to the Spanish population? Please describe this in a few short sentences.
  • Table 1 is difficult to read; possibly condense this to include most important information.
  • Figure 2 needs a bit more description, by itself, it does not improve the contents much, otherwise I would suggest not including figure 2.

Overall a good and relevant study. Please improve the presentation. Having all aims clearly stated in the beginning and then again in the conclusion makes it much easier to read.

Author Response

Response to Reviewer 1 Comments

The authors of the article entitled "Development of the Spanish version of Sniffin´s Sticks Olfactory Identification Test: normative data and validity of parallel measures" would like to thank you for your comments and suggestions that have contributed to substantially improve the quality of the manuscript submitted to the Brain Science journal. 

We refer to all the comments below.

  1. The aims discussed are not all mentioned in the abstract or the introduction, e.g. "the aim of this study was to obtain normative data for the group of people over 60" --> not mentioned in the abstract, this also applies to the "third objective"

Replay: Your comments and suggestions are appreciated very much. We have removed some content in the abstract, introduction, discussion and conclusions, we hope to have achieved a clearer explanation. The abstract and conclusions have been modified but the rest of the papers changes have been made that the reviewers will be able to see marked in yellow.

Abstract

            The Sniffin’ Sticks Olfactory Identification Test is a tool for measurement of olfactory performance developed in Germany and validated in several countries. This research aims to develop the Spanish version of the Sniffin´ Sticks Olfactory Identification Test and obtain normative values for the Spanish population. The parameters id free recall and subjective intensity of odorants are included. The influence of possible demographic covariates such as sex, age, smoking or educational level are analyzed, and the items that best discriminate are studied. In addition, the internal structure validity of the blue and purple versions is studied as a parallel measure, and the cultural adaptation of the purple version is carried out. For this three-independent samples of normosmic healthy volunteers were studied. To obtain normative values the sample was of 417 participants (18-89 years). For the internal structure validity study of both versions the sample was of 226 (18-70 years) and for familiarity of the purple version the sample was of 75 participants (21-79 years).  Results indicated that men and women, smokers and non-smokers perform equally. However, differences were found as age progresses, being more pronounced after 60 years old in all three measurements of the identification test. This research also provides the items that best discriminate in the blue version and the cultural adaptation for the purple version. In conclusion, the Sniffin Sticks Odor Identification Test is a suitable tool for olfactory assessment in Spanish population. The instrument has been expanded with two new scores and normative data as function of age are provided. Its parallel version seems also appropriate for testing, as items have been culturally adapted and evidence of internal structure validity for both versions is reported.

The goals at the end of the introduction

The goals of our study are: i) the development of normative data for the Spanish version of the Sniffin’ Sticks Olfactory Identification Test, including to the original test score (recognition score) two new scores (free recall and subjective intensity); ii) the analysis of covariables for each measurement and description of the normative values; iii) the study of the items that best discriminate across olfactory identification performance, so a potential shortened screening version of the Sniffin’ Sticks Olfactory Identification Test could be developed; iv) the obtention of evidences of the internal structure validity of the blue and purple versions of the test as parallel measures; v) the cultural adaptation of the purple version, by measuring the percentage of familiarity for each odor descriptor.

Conclusions

The present results obtained in this work constitute an important contribution in the evaluation of olfactory capacity, providing different normative data for each of the age groups.

The results do not indicate that there is a relationship between smell and sex, between smell and educational level, or between smell and smoking. However, changes in olfactory identification are observed as age progresses, changes are seen after 40 years old and the decrease being more pronounced after the age of 60,  in all three measures of identification capacity.

Other contributions of this research are the extraction of the items which best discriminate in the blue version of the test and that could be considered to be used as a shortened or screening version of the test. In addition, evidences of internal structure validity of both versions of the test (blue and purple) are provided through confirmatory factor analysis. Items from the purple version have also been adapted to Spanish population, as odor descriptors with lower percentages of familiarity were modified. Having a culturally-adapted, parallel version of the Sniffin Sticks Odor Identification Test supposes an important advantage in order to improve the quality of follow-up assessments.

In conclusion, the Sniffin’ Sticks Test is a suitable tool to evaluate olfactory identification ability within the clinical and research environment.

  1. Please adapt the abstract to clearly state the aims so that that is is well rounded (conclusion also doesn't fit the abstract)

Replay:  Your comments and suggestions are appreciated very much. We hope that we have achieved greater clarity in the presentation of the objectives in the abstract and conclusions, we hope you can appreciate it in the previous answer and in the article.

  1. In materials and methods you write that in any  case, the tip of the pen physically touched the participants nose. Could this be a miscommunication? Did you meant to write that in no case did it touch the nose?

Replay: Thank you for the observation. We have changed the sentence:

“In any case, the tip of the pen physically never touched the participant’s nose.”

  1. Please include more information on the free recall score in the methods (what exactly the possible scores meant)

Replay: Your comments and suggestions are appreciated very much.

In the free recall test, information is obtained from what the subject is able to remember spontaneously, without the help of clues (meaning the cards with the four possible answers as options assigned for the recognition task). When response alternatives are not offered (free recall), the cognitive load of the task is higher, and it offers us an added score of the individual's semantic memory capacity. Odor identification may be conceptualized as a semantic memory task in that it poses demands on an individual’s knowledge of a specific odor (Richardson, J. T. E., & Zucco, G. M. (1989). Cognition and olfaction: A review. Psychological Bulletin, 105, 352–360; Schab, F. R. (1991). Odor memory: Taking stock. Psychological Bulletin, 2, 242–251; Tulving, E. (1986). Episodic and semantic memory: Where should we go from here? Behavioral & Brain Sciences, 9, 573–577; Hedner, M.; Larsson, M. Arnold, N.; Gesualdo M. Zucco G.M. & Hummel, T. (2010) Cognitive factors in odor detection, odor discrimination, and odor identification tasks, Journal of Clinical and Experimental Neuropsychology, 32:10, 1062-1067).Studies indicate the existence of a positive relationship between general semantic memory and the identification of odors (Larsson, M., Finkel, D., & Pedersen, N. (2000). Odor identification: Influences of age, gender, cognition, and personality. Journals of Gerontology Series B: Psychological Sciences, 55, 304–310; Larsson, M., Nilsson, L., Olofsson, J. K., & Nordin, S. (2004). Demographic and cognitive predictors of cued odor identification: Evidence from a population-based study. ChemicalSenses, 29, 547–554).

The paragraphs have been modified following the suggestions made and the sentence:   

  1. Free recall score: As in a memory task, free recall implies the odor pen is presented and the participant has to guess the odor descriptor, without alternatives, doing their best to identify odor descriptor. In a free recall test, information is obtained from what a person is able to remember spontaneously, without the help of clues. This method also requires a major memory component, combined with smell identification. This score is obtained as the total of correct answers from the 16 items, when presented under free recall modality.

  1. How was the Odour identification test adapted to the Spanish population? Please describe this in a few short sentences.

Replay: Your comments and suggestions are appreciated very much. Odor Identification Test was adapted to the Spanish population by measuring the grade of familiarity with the odor descriptors (Delgado-Losada, M.L.; Delgado-Lima, A.H.; Bouhaben, J. (2020), https://doi.org/10.3390/brainsci10120943).

The paragraph has been modified following the suggestions made and the sentence:

Recognition score: This is the score proposed by the original version [58]. Odor Identification Test was adapted to the Spanish population by measuring the grade of familiarity with the odor descriptors [84]. The odor pen is presented to the participant and he has to recognize the target odor between four odor descriptors. Therefore, this score is obtained by a four-alternative forced-choice method. Correct answers from the 16 items are added in order to calculate this score.

  1. Table 1 is difficult to read; possibly condense this to include most important information.

Replay: Your comments and suggestions are appreciated very much. One of the aims of this study is to provide normative data of the Odor Identification Test in a large Spanish sample. Table 1 summarizes these efforts and it provides standardized values which might be used by other clinicians and researchers when they administer the instrument on Spanish population. Information included in this Table 1 is already the most important information in order to achieve this objective, as it provides statistics of average (mean and its confidence interval), scatter (standard deviation) and position (minimum, maximum and some percentiles). Categorization of this information as function of age is also necessary, as we report Age is a statistically significant covariate for each score. So, once again, your commentary is really appreciated, but we find condensing the information already shown impossible . Similar example of this kind of tables may be found in Oleszkiewicz et al., 2018 (doi.org/10.1007/s00405-018-5248-1).

  1. Figure 2 needs a bit more description, by itself, it does not improve the contents much, otherwise I would suggest not including figure 2.

Replay: Your comments and suggestions are appreciated very much. Legend of this figure has been changed to:

Figure 2. Confirmatory factor analysis models of Blue (A) and Purple (B) Odor Identification Tests. For each model, rectangles represent the items (empirical variables), ovals represent each factor. For each model, all items load into a unique factor, which could be named as odor identification construct. The standardized factorial load of every item is also reported in each model.

Overall a good and relevant study. Please improve the presentation. Having all aims clearly stated in the beginning and then again in the conclusion makes it much easier to read.

Replay: Sincerely, your comments and suggestions are appreciated very much. We hope we have improved the presentation, the clarity of the objectives, the abstract, the conclusions and the entire article in general.

Reviewer 2 Report

The paper by Delgado-Losada et al. entitled "Development of the Spanish version of Sniffin´s Sticks Olfac-2 tory Identification Test: normative data and validity of parallel 3 measures" have as objective to develop the Spanish version of Odor Identification Test and obtain normative values for the Spanish population, including the study of possible demographic covariates, such as sex, age or education level that could be related and influence the olfactory capacity. In general, I find the study interesting and its results useful for the scientific community and for clinical investigations in Spain. I am also pleased to point out that I greatly appreciated by the authors the self-criticism, which highlights the limitations of the study, but I believe that it is not enough to admit them and say "Future studies ....": there are some aspects that should be seriously revised.

Major

  • 1) It is not clear to which sample of the population the study is aimed. The authors say “The aim of this study was to obtain normative data for the group of people over 60 years of age…”, subsequently they argue “Our group is interested in analyzing deficits in semantic networks in future studies and studying the degree of olfactory…”. Although the authors point out "in future studies" this is confusing. I find that the standard parameters obtained on a sample of healthy people are not automatically translatable on patients with neurodegenerative diseases. Furthermore, if we wanted to accept this, the authors would have had to select a sample of people aged 50 and over, because they were more comparable with a certain type of patient. How many subjects for each age group? I believe it is an important aspect to verify the applicability of the results obtained. Indeed, if they wanted to obtain normalized data for the over 60s, why were younger subjects also selected?
  • 2) I do not share the concept that the study and evaluation of the olfactory function can be represented only by the ability to identify odors: I find it extremely reductive. I believe that the authors should re-evaluate this statement, which is repeated at various points in the manuscript.
  • 3) Not only do the authors think that it is sufficient to study the ability to identify odors to evaluate the olfactory function of subjects, but they even study the possibility of using a reduced test. I totally disagree.

Minor

Abstract

L10-11: to say that the Id-test is representative of olfactory sensitivity is a bit exaggerated: other skills are important that are not evaluated with the identification

L17: the great difference between males and females in numbers can be a problem. Women are more familiar with some smells and men with others.

L21: add space

L23: this does not seem to me a great discovery, so the authors should improve their conclusions.

Introduction

L35: the list is very long but it is not complete: autoimmune and inflammatory diseases are missing

L37: Aren't PD and AD already included in neurodegenerative diseases?

L44-45: I think the authors should better describe what is meant by olfactory threshold!

L62-63: similar results were also found in a study on Parkinon's disease patients (see Melis et al 2019)

L64: “internal validity…external validity” What does this mean? perhaps it should be explained better

L73-74: I can understand the time savings, but I honestly do not agree with the ris as a justification for using a reduced test.

L78: “he results suggest” I believe there is some error

L81: “Other studies, present” delete comma

L83-84: several studies argue the opposite, but the authors do not take it into account

L85-86: I do not agree with the authors that it is a little considered aspect. Several recent studies use scales for assessing the intensity perceived by subjects during the Id-test (Markovic et al 2007 Good news for elderly persons: olfactory pleasure increases at later stages of the life span; Fischer et al 2014 doi:10.1093/chemse/bju022; Sollai et al 2020 doi.org/10.1016/j.physbeh.2020.112820; Melis et al 2021 doi.org/10.1016/j.bbr.2021.113127)

L120-122: this objective seems to contradict what was previously said, that is the need to reduce the time, especially when dealing with patients. In fact, adding variants lengthens test delivery times.

M&M

L134, 134, 150: how old were they? were the samples homogeneous for sex and age?

L150: the self-assessment is known to be unreliable (vedi Steinbach et al 2013 doi:10.1371/

journal.pone.0073454)

L162-163: this concept should be written better

L168: I am a bit confused. But isn't this a goal of this paper?

L170-171: the authors should have reread the manuscript more carefully: is this sentence broken? missing parts? if not, it should be completely rewritten because it is not clear.

L180-181: this is true if the subjects give an assessment of intensity in relation to the reference value present in their memory. But their memory can be compromised, especially with age, and in various pathological conditions. Since the authors exclude some (clinically involved) subjects it may not have a real value.

L195: “both nostrils” together or separately?

L208-210: perhaps it would be better to move this sentence when it says that the tests were presented at a similar interval

L215: “local Ethic committee” protocol number and approval date should be provided

L222: what were these criteria?

L223-226: I think the authors should better explain these choices and analyzes. In fact, it turns out to be rather difficult to understand for those who do not use the same analyzes, associations or other.

L242: I find this procedure to have severe limitations. When you fill out online questionnaires you often do it casually, quickly, a little bored. So I wouldn't give these results much importance

L248: Again: it is known that self-assessments are never reliable. There is no relationship between the measured olfactory performance and the self-reported one

L249-250: i can't understand this explanation.

L266: here and later in the text the authors talk about "data cleaning", what does it mean?

L274: Why this choice?

L288: Why this value? In general, I think the authors should better explain the choice of cutoffs

L295-296: I believe that authors should not limit themselves to bibliographic references, but should provide more details in the text.

L309: why different characters?

Results

L328-329: This has already been said in M&M, I don't find it important to repeat it in the results

L339: Could this be due to the great inhomogeneity of the sample between women and men?

L393: There is one parenthesis too many

Table 4: I don't understand: why are there 50 "original odor descritpor" in the table, since the prurple identification test consists of 16 odors?

Delete the point after Gominola

Discussion

L406-407: I don't agree with this sentence: identification test can only evaluate one aspect of the olfactory function!

L419-424: these are results, so they should be moved to the right section! Also, is the 10th percentile value for differentiating normosmic from hyposmic the lower limit of normosmia or the upper limit of hyposmia?

L427-429: This concept is very interesting: it deserves to be developed better

L432-433: as shown by the high number of citations, the age effect is not so new, so the authors should give it less emphasis.

L428: “why, where and how"I disagree: having the reference values will allow the ezperimenter to assess whether the ability to identify odors in a population is normal or impaired.

L442-443: In reality there is a lot of literature that analyzes this aspect, not only linked to neurodegenerative diseases

L458-459: I guess the authors meant "especially from the effect of estrogen ..."

L470-472: I disagree: the smells identified in this study may not be really representative of the Spanish population: there could be differences related to regionality, experience, profession, type of pathology and duration, drug treatment, etc. I find it already quite reductive that the authors think they can evaluate the olfactory function only with the identification test, let alone if this is considerably reduced. I think the authors should review these aspects.

Conclusions

L529: in the first part of the discussion the authors highlight the beginning of the decay of the olfactory performance after the age of 40

L530-532: I do not agree with this conclusion, indeed I find this aspect limiting in the study of the olfactory function both in the clinical and research environment.

Author Response

Response to Reviewer 2 Comments

The authors of the article entitled "Development of the Spanish version of Sniffin´s Sticks Olfactory Identification Test: normative data and validity of parallel measures" would like to thank you for your comments and suggestions that have contributed to substantially improve the quality of the manuscript submitted to the Brain Science journal. 

We refer to all the comments below.

The paper by Delgado-Losada et al. entitled "Development of the Spanish version of Sniffin´s Sticks Olfactory Identification Test: normative data and validity of parallel measures" have as objective to develop the Spanish version of Odor Identification Test and obtain normative values for the Spanish population, including the study of possible demographic covariates, such as sex, age or education level that could be related and influence the olfactory capacity. In general, I find the study interesting and its results useful for the scientific community and for clinical investigations in Spain. I am also pleased to point out that I greatly appreciated by the authors the self-criticism, which highlights the limitations of the study, but I believe that it is not enough to admit them and say "Future studies ....": there are some aspects that should be seriously revised.

Major

  • 1) It is not clear to which sample of the population the study is aimed. The authors say “The aim of this study was to obtain normative data for the group of people over 60 years of age…”, subsequently they argue “Our group is interested in analyzing deficits in semantic networks in future studies and studying the degree of olfactory…”. Although the authors point out "in future studies" this is confusing. I find that the standard parameters obtained on a sample of healthy people are not automatically translatable on patients with neurodegenerative diseases. Furthermore, if we wanted to accept this, the authors would have had to select a sample of people aged 50 and over, because they were more comparable with a certain type of patient. How many subjects for each age group? I believe it is an important aspect to verify the applicability of the results obtained. Indeed, if they wanted to obtain normalized data for the over 60s, why were younger subjects also selected?

Replay: Your comments and suggestions are appreciated very much. We revised the complete paper and we hope we have improved all the aspects that were unclear. Perhaps we wanted to lead the way of our future studies and we have not been able to explain with clarity. We hope that we have now conveyed that necessary clarity. We sincerely appreciate all of your contributions for they have been useful in improving this paper.

2) I do not share the concept that the study and evaluation of the olfactory function can be represented only by the ability to identify odors: I find it extremely reductive. I believe that the authors should re-evaluate this statement, which is repeated at various points in the manuscript.

Replay: Your comments and suggestions are appreciated very much. You are right in saying that this statement might be reductionist, we have removed it in our discussion. However, in the introduction we thought it convenient to leave the sentence due to previous work in that line. (Several studies indicate that the Sniffin’ Stick Olfactory Identification Test alone may function as a screening test for olfactory dysfunction or follow-up of olfactory function [85,86]  85. Jackman, A. H.; Doty, R. L. Utility of a three-item smell identification test in detecting olfactory dysfunction. https://doi.org/10.1097/01.mlg.0000183194.17484.bb;  86. Negoias, S.; Troeger, C.; Rombaux, P.; Halewyck, S.; Hummel, T.; Number of descriptors in cued odor identification tests. https://doi.org/10.1001/archoto.2009.231).

In both, discussion and conclusion sections, we have indicated the usefulness of the usage of the odorants that best discriminate for the detection of patients who require a more complete and exhaustive evaluation of their olfactory capacity.

3) Not only do the authors think that it is sufficient to study the ability to identify odors to evaluate the olfactory function of subjects, but they even study the possibility of using a reduced test. I totally disagree.

Replay: Your comments and suggestions are appreciated very much. The odor identification test is a complementary tool for evaluation of the olfactory capacity. This is how we wanted to convey it and perhaps it was not well expressed. We have corrected it in the manuscript, not as a determined test in the assessment, but as a valid and reliable olfactory assessment instrument as demonstrated in a large number of studies [61-85].

Minor

Abstract          The abstract was changed substantially.

L10-11: to say that the Id-test is representative of olfactory sensitivity is a bit exaggerated: other skills are important that are not evaluated with the identification

Replay: Your comments and suggestions are appreciated very much. What we mean by that is that is one measurement of the olfactory performance, a complementary measure of the total olfactory capacity.

L17: the great difference between males and females in numbers can be a problem. Women are more familiar with some smells and men with others.

Replay: We agree on your appreciation. The high proportion of female participants might be masking potential differences. However, gender differences are far from consensus. Regarding this aspect, discrepancies between several studies have been found (Sorokowski et al., 2019 doi.org/10.3389/fpsyg.2019.00242; Brand & Millot, 2001 DOI: 10.1080/02724990143000045; Hummel et al., 2003 doi.org/10.1016/S0378-4274(03)00078-X; Corwin et al., 1995 doi.org/10.1093/geronb/50B.4.P179; Doty & Cameron, 2009 doi.org/10.1016/j.physbeh.2009.02.032; Larson et al., 2003 doi.org/10.1016/S0001-6918(02)00092-6; Oberg et al., 2002 doi.org/10.1017/S1355617702801424). Our results seem to point to a lack of gender differences in olfactory performance.

L21: add space

Replay: Thanks for the correction.

L23: this does not seem to me a great discovery, so the authors should improve their conclusions.

 Replay: Your comments and suggestions are appreciated very much. We have removed some content in the abstract, introduction, discussion and conclusions, we hope to have achieved a clearer explanation. The abstract and conclusions have been modified but the rest of the papers changes have been made that the reviewer will be able to see marked in yellow.

Abstract

The Sniffin’ Sticks Olfactory Identification Test is a tool for measurement of olfactory performance developed in Germany and validated in several countries. This research aims to develop the Spanish version of the Sniffin´ Sticks Olfactory Identification Test and obtain normative values for the Spanish population. The parameters id free recall and subjective intensity of odorants are included. The influence of possible demographic covariates such as sex, age, smoking or educational level are analyzed, and the items that best discriminate are studied. In addition, the internal structure validity of the blue and purple versions is studied as a parallel measure, and the cultural adaptation of the purple version is carried out. For this three-independent samples of normosmic healthy volunteers were studied. To obtain normative values the sample was of 417 participants (18-89 years). For the internal structure validity study of both versions the sample was of 226 (18-70 years) and for familiarity of the purple version the sample was of 75 participants (21-79 years).  Results indicated that men and women, smokers and non-smokers perform equally. However, differences were found as age progresses, being more pronounced after 60 years old in all three measurements of the identification test. This research also provides the items that best discriminate in the blue version and the cultural adaptation for the purple version. In conclusion, the Sniffin Sticks Odor Identification Test is a suitable tool for olfactory assessment in Spanish population. The instrument has been expanded with two new scores and normative data as function of age are provided. Its parallel version seems also appropriate for testing, as items have been culturally adapted and evidence of internal structure validity for both versions is reported.

Conclusions

The present results obtained in this work constitute an important contribution in the evaluation of olfactory capacity, providing different normative data for each of the age groups.

The results do not indicate that there is a relationship between smell and sex, between smell and educational level, or between smell and smoking. However, changes in olfactory identification are observed as age progresses, changes are seen after 40 years old and the decrease being more pronounced after the age of 60,  in all three measures of identification capacity.

Other contributions of this research are the extraction of the items which best discriminate in the blue version of the test and that could be considered to be used as a shortened or screening version of the test. In addition, evidences of internal structure validity of both versions of the test (blue and purple) are provided through confirmatory factor analysis. Items from the purple version have also been adapted to Spanish population, as odor descriptors with lower percentages of familiarity were modified. Having a culturally-adapted, parallel version of the Sniffin Sticks Odor Identification Test supposes an important advantage in order to improve the quality of follow-up assessments.

In conclusion, the Sniffin’ Sticks Test is a suitable tool to evaluate olfactory identification ability within the clinical and research environment.

Introduction

L35: the list is very long but it is not complete: autoimmune and inflammatory diseases are missing

Replay: Your comments and suggestions are appreciated very much. The paragraph has been modified following the suggestions made and the sentence:

“rhinitis, sinusitis [11,12], autoimmune diseases [13,14], inflammatory diseases [15] anxiety [16,17],”

L37: Aren't PD and AD already included in neurodegenerative diseases?

Replay: Your comments and suggestions are appreciated very much. Yes, we included them as an example... neurodegenerative diseases [24,25], such as frontotemporal dementia [26-28] amyotrophic lateral sclerosis [29] Parkinson's disease [30-33], or Alzheimer's disease [33-36].

L44-45: I think the authors should better describe what is meant by olfactory threshold!

Replay: Your comments and suggestions are appreciated very much. We have changed the sentence indicated by the reviewer:

The olfactory threshold represents the level of odour detection at low concentration meaning the least detectable concentrations of odorant that can be perceived…

L62-63: similar results were also found in a study on Parkison's disease patients (see Melis et al 2019)

Replay:  Your comments and suggestions are appreciated very much. We revised the paper and thought it was of great interest for the discovery of a specific polymorphism in the gene coding the olfactory function, related with Parkinson's disease in women. Thank you very much for the support.

In addition, we edited the line 62-63, as follows:

Several studies indicate that the Sniffin’ Stick Olfactory Identification Test alone may function as a screening test for olfactory dysfunction or follow-up of olfactory function [85,86], and they are more feasible to apply in clinical practice [68,69].

L64: “internal validity…external validity” What does this mean? perhaps it should be explained better

Replay: Your comments and suggestions are appreciated very much. We have changed the sentence indicated by the reviewer. The paragraphs have been modified following the suggestions made and the sentence:

Several evidences of test validity have been obtained in other cultures and languages [58,61,72,73,77,78].

L73-74: I can understand the time savings, but I honestly do not agree with the ris as a justification for using a reduced test.

Replay:  Your comments and suggestions are appreciated very much. We thought it was convenient to eliminate the sentence (reducing the time and cost of the personnel administering the test). The paragraph has been modified the sentence:

... for example, on the number of odorants used, mostly concluding the usefulness of the reduced versions in clinical practice [70,86-88].

L78: “he results suggest” I believe there is some error.

Replay: Thank you for the observation. We have changed the sentence:

“… versions, the results suggest that odor identification …”

L81: “Other studies, present” delete comma

Replay: Thank you for the observation. We have changed the sentence:

“Other studies present modifications …”

L83-84: several studies argue the opposite, but the authors do not take it into account

Replay:  Your comments and suggestions are appreciated very much. We have included one study in the lines that the reviewer indicates (Ute Walliczek-Dworschak, U; Pellegrino, R.; Shangwa, L.; Hummel, C.; Antje, H, and Hummel, T. (2016)). The paragraph has been modified following the suggestions made and the sentence:

Other studies present modifications of the odorants presentation order and the type of labels (verbal descriptors with or without pictures) finding that scores were not significantly different when the subjects were presented either with verbal descriptors only or with verbal descriptors and pictures [92], but differences were found in performance when including background noise or positive concurrent feedback  [93].

L85-86: I do not agree with the authors that it is a little considered aspect. Several recent studies use scales for assessing the intensity perceived by subjects during the Id-test (Markovic et al 2007 Good news for elderly persons: olfactory pleasure increases at later stages of the life span; Fischer et al 2014 doi:10.1093/chemse/bju022; Sollai et al 2020 doi.org/10.1016/j.physbeh.2020.112820; Melis et al 2021 doi.org/10.1016/j.bbr.2021.113127)

Replay: Your comments and suggestions are appreciated very much. We included the bibliographic references provided by the reviewer. The paragraph has been modified following the suggestions made and the sentence:

An aspect of great interest, considered by a few studies in which the Sniffin´ Stick Olfactory Identification Test has been used is the perceived subjective intensity with which each odorant is perceived [94-96].

L120-122: this objective seems to contradict what was previously said, that is the need to reduce the time, especially when dealing with patients. In fact, adding variants lengthens test delivery times.

Replay: Your comments and suggestions are appreciated very much. We agree with you that it might seem contradictory, but our objective is to analyze additional parameters in the usage of the identification test that would be of more interest in the scientific field, and, in parallel, to study which odorants best discriminate and could constitute the elements of choice in the event that the application time was brief, it would be of more interest in the clinical context a fact that we know from the group´s relationship with the otorhinolaryngology service of the Hospital Clinico San Carlos de Madrid, as well as other similar medical centers.

M&M

L134, 134, 150: how old were they? were the samples homogeneous for sex and age?

Replay: Your comments and suggestions are appreciated very much. We have changed the sentence indicated by the reviewer. The paragraphs have been modified following the suggestions made and the sentence:

Both versions of the Odor Identification Test (Blue and Purple versions) were administered to a second sample of 235 participants (final sample 134 of n = 226), aged between 18-79.

A third independent sample of 83 participants (final sample of n = 75), aged between 21-79, was asked to fulfill a questionnaire about the familiarity with the odor descriptors from Odor Identification Test (Purple version).

In L 232:

A sample of 226 participants, aged between 18-79 (mean age = 49.66, SD = 15.03) (…).

In L 241, L 242

(…) in a sample of 75 participants, aged between 21-79 (mean age = 51.83, SD = 21.23).

L150: the self-assessment is known to be unreliable (vedi Steinbach et al 2013 doi:10.1371/ journal.pone.0073454)

Replay: Your comments and suggestions are appreciated very much. Study 3 does not measure olfactory function, but familiarity with certain odor descriptors. Hence, self-assessment is the most parsimonious way to measure the grade of familiarity with certain odor descriptors.

In addition, the clinical use of self-report questionnaires plays a central role in the practice of clinical psychology. The psychometric properties of self-assessment suggest that it is a valid and reliable instrument for information collection, as indicated in the properties of questionnaires widely used as Beck Depression Inventory (Dozois, D. J. A.; Dobson, K. S.; Ahnberg, J. L. A psychometric evaluation of the Beck Depression Inventory–II. Psychological Assessment. 1998, 10(2), 83–89. https://doi.org/10.1037/1040-3590.10.2.83)

L162-163: this concept should be written better.

Replay:  Your comments and suggestions are appreciated very much. We have changed the sentence indicated by the reviewer. The paragraph has been modified following the suggestions made and the sentence:

                                                                                                  The chance of measuring the same construct using parallel versions of the same measurement instrument has a clear advantage: retesting allows control for practice effect.

L168: I am a bit confused. But isn't this a goal of this paper?

Replay: Your comments and suggestions are appreciated very much. We have changed the sentence indicated by the reviewer. The paragraph has been modified following the suggestions made and the sentence:

Free recall score: As in a memory task, free recall implies the odor pen is presented and the participant has to guess the odor descriptor, without alternatives, doing their best to identify the odor descriptor.

L170-171: the authors should have reread the manuscript more carefully: is this sentence broken? missing parts? if not, it should be completely rewritten because it is not clear. Thank you, alterations were made

Replay:  Your comments and suggestions are appreciated very much. Although we have already reread the manuscript and the sentence was not broken, we have changed it for a better understanding. The paragraph has been modified following the suggestions made and the sentence:

This score is obtained as the total of correct answers from the 16 items, when presented under free recall modality.

L180-181: this is true if the subjects give an assessment of intensity in relation to the reference value present in their memory. But their memory can be compromised, especially with age, and in various pathological conditions. Since the authors exclude some (clinically involved) subjects it may not have a real value.

Replay:  Your comments and suggestions are appreciated very much. The intention of the subjective intensity assessment is to provide a value, from 0 to 10, which represents how intense is the smell for the participant. As any other subjective measure, it could be biased by multiple factors. However, we provide data on how these subjective values are also affected, not only by the individual experience of smelling, but also by group variables such as age. We intend to provide normative data (that is why clinical subjects are out of the normative sample) on how people tend to experience the intensity of smelling depending on their ages. Trend shows that the subjectively perceived intensity of odors decreases with age, this is our results even though it is contradictory to what was found t¡in the study of Markovic et al., 2007, provided by the reviewer. The mechanisms underlying this decrease are out of the scope of the paper, but memory dependency and its affection with age is a very interesting hypothesis for future studies.

L195: “both nostrils” together or separately?

Replay: Your comments and suggestions are appreciated very much. We have changed the sentence indicated by the reviewer. The paragraph has been modified following the suggestions made and the sentence:

Olfactory function was assessed for both nostrils, together.

L208-210: perhaps it would be better to move this sentence when it says that the tests were presented at a similar interval.

Replay: Your comments and suggestions are appreciated very much. We have changed the sentence indicated by the reviewer. The paragraph has been modified following the suggestions made and the sentence:

Both Blue and Purple versions of the odor identification test were administered to the same participants in a short and similar time interval, by the same evaluator under the same environmental situation. The two versions were presented in two different sessions, with an interval between them of 7 to 10 days. Both versions are similar in content, format and instructions: same number, type, difficulty and time of application of the odorant pens. A counterbalance was made in terms of the order of presentation, that is, while approximately half of participants began with the blue version, the other half began with the purple version. The allocation of participants to both groups was random.

L215: “local Ethic committee” protocol number and approval date should be provided

Replay: Your comments and suggestions are appreciated very much. We have already sended the information via email to the editor, we are sorry that it wasn't included before. We have changed the sentence indicated by the reviewer. The paragraph has been modified following the suggestions made and the sentence:

The study was ruled by the principles of the Declaration of Helsinki (Edinburgh, 2013) and was approved by the Ethics Committee from University Hospital San Carlos (Madrid, Spain) (ref. number: 17/192-E).

L222: what were these criteria?

 Replay: Your comments and suggestions are appreciated very much. Eligibility criteria is presented in the Participants section. The paragraph has been modified following the suggestions made and the sentence:

Due to eligibility criteria (see section 2.1 Participants), from this initial sample, final sample was (…).

L223-226: I think the authors should better explain these choices and analyzes. In fact, it turns out to be rather difficult to understand for those who do not use the same analyzes, associations or other.

Replay: Your comments and suggestions are appreciated very much. We have changed the sentence indicated by the reviewer. The paragraph has been modified following the suggestions made and the sentence:

Due to eligibility criteria (see section 2.1 Participants), from this initial sample, the final sample was composed by 417 participants (291 females and 125 males) aged from 20 to 84 years (mean age = 58.94, SD = 13.73). Normative data was obtained for this sample (statistics of average, scatter and position) and item analysis (difficulty index/mean score per item, biserial correlation and corrected point-biserial correlation as discrimination index) was performed in order to check the quality of the items.

L242: I find this procedure to have severe limitations. When you fill out online questionnaires you often do it casually, quickly, a little bored. So I wouldn't give these results much importance.

Replay: Your comments and suggestions are appreciated very much. Boring, quickness and/or reluctance are serious issues which are not limited to online questionnaires. Typical performance, self-report questionnaires might depend on the predisposition of the participant to fulfill them (Fernandez-Ballesteros, 2007 ISBN: 8436825489), whether the administration is online or by pencil and paper. Factors such as quickness may not affect questionnaire reliability (Montag & Reuter, 2008; doi.org/10.1089/cpb.2007.0258). Besides, this methodology was previously presented in Delgado-Losada et al., (2020).

L248: Again: it is known that self-assessments are never reliable. There is no relationship between the measured olfactory performance and the self-reported one.

Reply: Your comments and suggestions are appreciated very much. As this secondary sample was recruited online, due to, among other reasons, the situation of world pandemic, we faced with the necessity to establish a minimum olfactory function criteria. We established this one as we consider that it represents the minimum point for a positive perception of olfactory function.

L249-250: i can't understand this explanation.

Replay:  Your comments and suggestions are appreciated very much. As it was stated in the previous comment, we considered including in the familiarity sample participants who consider themselves to have a positive perception of their olfactory function. In a scale from 0 to 10 (as in scholar grades), the point which represents a positive perfection, rather than a negative one, is 5.

L266: here and later in the text the authors talk about "data cleaning", what does it mean?

Reply: Your comments and suggestions are appreciated very much. Data cleaning means the process to detect, correct or remove wrong or inaccurate data records in a database. In the case of the present study, it refers to the detection of wrong records and/or participants who do not comply with eligibility data are removed from the database. The paragraph has been modified following the suggestions made and the sentence:

After outlier detection and data cleaning (detection and removal of wrong records and records from participants who do not comply with eligibility criteria) (…).

L274: Why this choice?

Reply: Your comments and suggestions are appreciated very much. Stepwise method was chosen as it is a method that allows more flexibility to examine multiple models for exploratory purposes.

L288: Why this value? In general, I think the authors should better explain the choice of cutoffs

Replay: Your comments and suggestions are appreciated very much. References 98 and 99 cover the choice of this cutoff points. We have improved the sentence indicated by the reviewer. The paragraph has been modified following the suggestions made and the sentence:

Corrected point-biserial correlation is interpreted as the discrimination index, as how much the item discriminates between participants’ odor identification performance (i.e. if good smellers or participants with higher olfactory performance are more likely to score the item than participants with lower olfactory performance or bad smellers, it is said that the item is a good discriminant of olfactory function). Cutoff point in this index is traditionally set at 0.2 [105,106]. Items whose discrimination index is below 0.2 are considered to be checked. Items with discrimination index equal or greater than 0.2 are acceptable, and those equal or greater than 0.3 (but lower than 0.7) are good discriminant items. Cronbach’s alpha was calculated for each score.

L295-296: I believe that authors should not limit themselves to bibliographic references, but should provide more details in the text.

Reply: Your comments and suggestions are appreciated very much. From our point of view, explanation of robust weighted least squares as extraction method is out of the scope of a study about olfactory function. The reasons why we chose this method are stated and a reference which goes deeper into the statistical background of robust weighted least squares method is provided. However, we have improved the sentence indicated by the reviewer. The paragraph has been modified following the suggestions made and the sentence:

Robust weighted least squares was picked as parameter estimation method, as the traditionally chosen maximum likelihood method supposes continuous empirical variables adjusted to a multivariate normal distribution, which is not the case. This method was chosen because it uses tetrachoric correlation for factor extraction (see Flora & Curran [107] for an in-depth explanation of WLSMV method and its advantages versus maximum likelihood method with dichotomous empirical variables).

L309: why different characters?

Reply: Your comments and suggestions are appreciated very much. It is a typographic mistake. The paragraph has been modified following the suggestions made and the sentence:

(Tucker-Lewis Index) and CFI (Comparative Fit Index) were also considered.

Results

L328-329: This has already been said in M&M, I don't find it important to repeat it in the results

Reply: Your comments and suggestions are appreciated very much. We have changed the sentence indicated by the reviewer. The paragraph has been modified following the suggestions made and the sentence:

Descriptive analysis was performed over the three odor identification scores. Descriptive statistics from this normative sample (Odor Identification Test, Blue version) are shown in Table 1.

L339: Could this be due to the great inhomogeneity of the sample between women and men?

Replay: Your comments and suggestions are appreciated very much. We have added to the Discussion section (L459) a comment on how the high percentage of female participants in this study might mask potential gender differences:

Although others indicate that women perform better in the olfactory test due to hormonal factors, especially estrogens in the female olfactory epithelium [60,78,86]. The high proportion of female participants in our sample might also mask potential gender differences. Thus, descriptive statistics per sex and age are attached to Supplementary Material (Table S1).

L393: There is one parenthesis too many years.

Replay: Your comments and suggestions are appreciated very much. There is a nested parentheses inside the parenthesis.

Table 4: I don't understand: why are there 50 "original odor descriptor" in the table, since the purple identification test consists of 16 odors?

Replay: Your comments and suggestions are appreciated very much. Purple Identification Test consists of 16 four-choices (one odor target and three distractors) items, giving a total of 16 x 4 = 64 odor descriptors. From these 64 odor descriptors, there are 50 different odor descriptors (some items have repeated odor descriptors as distractors).

Delete the point after Gominola

Replay:  Thank you for the observation. We have removed the point:  “Gominola”

Discussion

L406-407: I don't agree with this sentence: identification test can only evaluate one aspect of the olfactory function!

Replay:  Your comments and suggestions are appreciated very much.  You are right, that we have overstated the statement. The paragraph has been modified following the suggestions made and the sentence:

The Sniffin’ Stick Olfactory Identification Test is a screening test for olfactory dysfunction or follow-up of olfactory function clinically [68,69,85,86].

L419-424: these are results, so they should be moved to the right section! Also, is the 10th percentile value for differentiating normosmic from hyposmic the lower limit of normosmia or the upper limit of hyposmia?

Replay: Your comments and suggestions are appreciated very much. We have modified the sentence indicated by the reviewer. Regarding the 10th percentile, it has been used as cutoff point for discriminating hyposmia from normosmia, as stated by Hummel et al., 2007 (doi: 10.1007/s00405-006-0173-0)

The paragraph has been modified following the suggestions made and the sentence:

The normative data presented in tables to be used as a guide to estimate individual olfactory identification capacity in relation to the individual’s age. The normative data of the three scores that make up the validation of the Spanish version of the identification test are: Free recall, Recognition and Intensity. The tables allow us to compare the performance of people over 20 years old, assigning a range of deciles compared to their peers of a similar age. The decision about this age categorization by 10 years was made based on the intention to capture olfactory differences across the lifespan, following the same procedures studies in the area including our previous work [60,74,84]. The 10th percentile has been used to discriminate between normosmic and hyposmic people [58].

L427-429: This concept is very interesting: it deserves to be developed better

Replay: Thank you very much for your interest. The Dunning-Kruger effect is known as a metacognitive illusory in individuals who tend to believe they perform better than others. This effect has been studied in young people who may self-perceive that they perform better than they actually do. The original account of the Dunning-Kruger effect involves the idea that metacognitive insight requires the same skills as task performance, so that unskilled performed poorly and lack of insight. However, global measures of self-assessment are prone to statistical and other biases that could explain the same pattern.

You can see:

McIntosh, R. D., Fowler, E. A., Lyu, T., & Della Sala, S. (2019). Wise up: Clarifying the role of metacognition in the Dunning-Kruger effect. Journal of Experimental Psychology: General, 148(11), 1882–1897. https://doi.org/10.1037/xge0000579

Muller, A., Sirianni, L. A., & Addante, R. J. (2020). Neural correlates of the Dunning–Kruger effect. European Journal of Neuroscience.  https://doi.org/10.1111/ejn.14935

Dunning, D. (2011). The Dunning–Kruger effect: On being ignorant of one's own ignorance. In Advances in experimental social psychology (Vol. 44, pp. 247-296). Academic Press.

L432-433: as shown by the high number of citations, the age effect is not so new, so the authors should give it less emphasis.

Replay:  Your comments and suggestions are appreciated very much. We have changed the sentence indicated by the reviewer, we have deleted some references. The paragraph has been modified following the suggestions made and the sentence:

… This decrease in the ability to identify odors related to the aging process has been described in numerous previous studies [37-39].

L428: “why, where and how"I disagree: having the reference values will allow the ezperimenter to assess whether the ability to identify odors in a population is normal or impaired. (438)

Replay: Your comments and suggestions are appreciated very much. It was a mistake on our part to express the usefulness of having normative values in this way. We hope that the normative data is used as a tool allowing the clinics on their daily practice a evaluation more precisely. The paragraph has been modified following the suggestions made and the sentence:

One of the objectives of the current study was to develop the Spanish version of the Sniffin Sticks Odor Identification Test and to obtain normative data of this population. Within this objective, we give special relevance to the +60 cohort, as, from these results, we could plan future studies which dive deeper in the odor identification performance for these ages. Having reference values of the identification test with the Free recall, Recognition and Intensity measures will allow an assessment of whether the ability to identify odours in a population is normal or impaired. It might be useful to have normative values for each parameter.

L442-443: In reality there is a lot of literature that analyzes this aspect, not only linked to neurodegenerative diseases

Replay: Your comments and suggestions are appreciated very much. we are in agreement with the reviewer and proceeded to eliminate the sentence... Althoug its loss is well documented, little has been studied about the basis for its age-related decline.... The paragraph has been modified following the suggestions made and the sentence:

… Olfactory identification requires semantic knowledge, the ability to retrieve it and to associate the smell retrieved from memory with a linguistic tag. Difficulties at any level of semantic processing can disrupt task performance. Although the deficit in the organization of semantic knowledge in patients with Alzheimer´s disease in known, the hypothesis of a break in the semantic network for odors is suggested [113-115].

L458-459: I guess the authors meant "especially from the effect of estrogen ..."

Replay:  Your comments and suggestions are appreciated very much. Yes, corrected.  The paragraph has been modified following the suggestions made and the sentence:

… Although others indicate that women perform better in the olfactory test due to hormonal factors, especially from the effect of estrogens in the female olfactory epithelium [60,78,86].

L470-472: I disagree: the smells identified in this study may not be really representative of the Spanish population: there could be differences related to regionality, experience, profession, type of pathology and duration, drug treatment, etc. I find it already quite reductive that the authors think they can evaluate the olfactory function only with the identification test, let alone if this is considerably reduced. I think the authors should review these aspects.

Replay: Your comments and suggestions are appreciated very much. We have deleted that sentence. The paragraph has been modified following the suggestions made and the sentence:

These seven odorants could constitute the abbreviated version of the blue test of the Spanish version of the Sniffin´ Stick Olfactory Identification Test, in the same line of the abbreviated versions proposed by other authors [68,70,88] that could be useful identifying patients who should undergo a more exhaustive and extensive evaluations of their olfactory capacity.

Conclusions

L529: in the first part of the discussion the authors highlight the beginning of the decay of the olfactory performance after the age of 40

Replay:  Your comments and suggestions are appreciated very much. The paragraph has been modified following the suggestions made and the sentence:

However, changes in olfactory identification are observed as age progresses, changes are seen after 40 years old and the decrease being more pronounced after the age of 60,  in all three measures of identification capacity.

L530-532: I do not agree with this conclusion, indeed I find this aspect limiting in the study of the olfactory function both in the clinical and research environment.

Replay: Your comments and suggestions are appreciated very much. We have modified the conclusion of the study, as well as other sections. We have indicated that the possibility of reducing the number of odorants, using those that best discriminate could constitute a measure to identify patients who should undergo a more exhaustive and extensive evaluations. We deeply appreciate all of your contributions because it allowed us to improve the quality of our manuscript.

… in the same line of the abbreviated versions proposed by other authors [68,70,88] that could be useful identifying patients who should undergo a more exhaustive and extensive evaluations of their olfactory capacity.

Round 2

Reviewer 2 Report

I congratulate the authors on their work and on improving their manuscript